# Study on Crystallization Process of Li_2_O–Al_2_O_3_–SiO_2_ Glass-Ceramics Based on In Situ Analysis

**DOI:** 10.3390/ma15228006

**Published:** 2022-11-12

**Authors:** Minghan Li, Chunrong Xiong, Yanping Ma, Hong Jiang

**Affiliations:** 1State Key Laboratory of Marine Resource Utilization in South China Sea, Special Glass Key Laboratory of Hainan Province, Hainan University, Haikou 570228, China; 2Key Laboratory of Advanced Materials of Tropical Island Resources of Ministry of Education, Haikou 570228, China

**Keywords:** crystallization study, in situ analysis, crystal growth, Li_2_O–Al_2_O_3_–SiO_2_ glass-ceramics

## Abstract

In this paper, we used differential scanning calorimetry (DSC), high-temperature X-ray diffraction (HT-XRD), and confocal scanning laser microscopy (CSLM) to investigate the Li_2_O–Al_2_O_3–_SiO_2_ glass crystallization process. At 943 K, lithium disilicate (Li_2_Si_2_O_5_) phase crystals began to precipitate in the Li_2_O–Al_2_O_3_–SiO_2_ glass with a crystal size of 50–70 nm. At the temperature of 1009 K, petalite (LiAlSi_4_O_10_) crystals began to precipitate in the vitreous phase, forming composite spherical crystals of LiAlSi_4_O_10_ and Li_2_Si_2_O_5_ with size in the range of 90–130 nm. Furthermore, the Kissinger method and KAS method of the JMAK model were used to calculate the crystallization activation energy and the Avrami index “*n*”. It was found that the precipitation mechanism of the two kinds of crystals is whole crystallization; accordingly, the selection of crystallization heat treatment system was guided to determine the nucleation and crystallization temperature.

## 1. Introduction

Glass-ceramics are materials with the characteristics of ceramic crystals, featuring not only high mechanical strength and hardness, but also good wear resistance [1]. They can also achieve special optical, electrical, magnetic, thermal, and biological functions through glass composition design, making them widely used as technical materials, structural materials, or other functional materials. Li_2_O–Al_2_O_3_–SiO_2_ glass-ceramics are among the most widely studied glass-ceramics, due to their ultralow thermal expansion coefficient, high transparency, and better chemical durability and mechanical properties than high-aluminosilicate glass [2]. By controlling the type and volume fraction of crystallization, its thermal expansion coefficient can be adjusted within a wide range, whereby even zero expansion or negative expansion can be realized, thus meeting different use requirements [1,2]. The Vickers hardness and bending strength could reach 900 and 100 MPa, respectively, exceeding the mechanical properties of high-aluminosilicate glass (Vickers hardness: 750–800; bending strength: 80–90 MPa) [3]; therefore, it has important commercial value in touchscreens, induction cooker panels, and other fields [4,5].

Generally, the crystalline phases precipitated from Li_2_O–Al_2_O_3_–SiO_2_ glass were mainly β-quartz solid solution and β-spodumene solid solution [6]. In addition, glass-ceramics with petalite, lithium disilicate, and hydrothermal quartz as the main crystalline phases appeared. The type, volume fraction, and morphology of the microcrystalline phase in glass had a great influence on the mechanical, optical, thermal, and electrical properties of glass. The β-quartz and β-spodumene solid solution had small thermal expansion coefficients [7,8]; however, the crystallization temperature range was narrow, making it difficult to control the crystal size. Petalite and lithium disilicate crystals had excellent mechanical properties [9,10], but increasing the number of these crystals in the glass phase and controlling the grain size represent challenges in current research. The crystal size, volume fraction, and morphology are closely related to nucleation and crystal growth. Therefore, it is necessary to study the crystallization process of glass, master the crystallization mechanism, control the appropriate nucleation temperature and holding time, and provide guidance for the preparation of ideal glass-ceramics.

The common research methods of crystallization kinetics are isothermal methods and non-isothermal methods [11,12,13,14,15]. Isothermal methods study the influence of time and temperature on the process and rate of crystal formation under constant temperature. However, rigorous isothermal experiments are actually difficult to achieve, especially at the initial stage of reaction. In non-isothermal methods, the thermal analysis curve contains the information of several isothermal curves, making the analysis faster, simpler, and more accurate [16]. Kumar et al. analyzed the crystallization kinetics of Li_2_O–Al_2_O_3_–SiO_2_ glass using a linear model, nonlinear model, and model-free non-isothermal method [17], optimized the crystallization heat treatment processing, and prepared highly transparent glass-ceramics with an extremely low thermal expansion coefficient. Nevertheless, the kinetic analysis could not reflect the change in kinetic parameters in the crystallization process, and the specific crystallization process could not be mastered. Huidong Li et al. observed the crystallization process of Li_2_O–Al_2_O_3_–SiO_2_ glass using a high-temperature confocal laser scanning microscope and calculated the crystallization activation energy on the basis of the results of differential thermal analysis [18]. The nucleation and growth process of crystals could be directly observed using the high-temperature confocal laser scanning microscope, but the non-isothermal model selected could not reflect the specific crystallization process or be analyzed in combination with the in situ test results. Lola Lilensten et al. [19] used an isothermal kinetic model to analyze the high-temperature X-ray diffraction of Li_2_O–Al_2_O_3_–SiO_2_ glass; however, due to the limitations of isothermal analysis, the results were unable to accurately guide the heat treatment process.

In this paper, two non-isothermal models are used to analyze the crystallization process of Li_2_O–Al_2_O_3_–SiO_2_ glass, and the crystallization kinetics parameters and mechanism are analyzed. High-temperature X-ray diffraction (HT-XRD) and confocal scanning laser microscopy (CSLM) are used for in situ testing of Li_2_O–Al_2_O_3_–SiO_2_ glass. Compared with the research results of Kumar, Huidong Li, and Lilensten [18,19,20], a non-isothermal model is applied to the analysis of HTXRD test results, thereby reducing the limitations of the isothermal model in the actual crystallization process, and allowing a more accurate analysis of the real-time crystallization kinetic parameters, such that the crystallization kinetic analysis is more accurate. Confocal scanning laser microscopy (CSLM), which unifies laser, electronic camera, and computer image processing technology, provides the possibility for continuous real-time observation of crystal number, morphology, and size at high temperatures [18]. Through real-time observation of the crystallization image of Li_2_O–Al_2_O_3_–SiO_2_ glass, it is convenient to analyze the relationship linking crystal growth rate, grain number, and grain size.

## 2. Experiment

A compound was prepared with the following composition (wt.%): 66.09 SiO_2_, 4.91 Al_2_O_3_, 1.5 Na_2_O, 20.00 Li_2_O, 2.03 B_2_O_3_, 1.22 P_2_O_5_, 2.25 ZrO_2_, and 2 ZnO_2_. The batch was composed of SiO_2_, H_3_BO_3_, and NH_4_H_2_PO_4_ (purity: 98%; XiLONG SCIENTIFIC, Guangzhou, China), Li_2_CO_3_, Na_2_CO_3_, and Al_2_O_3_ (purity: 99%; Macklin, Shanghai, China), and ZrO_2_ and ZnO_2_ (purity: 99%; Alladdin, Shanghai, China).

The addition of Na_2_O and B_2_O_3_ can reduce the viscosity of the molten glass at high temperature and play the role of flux. Adding small amounts of P_2_O_5_ and ZrO_2_ played the role of a nucleating agent, whereby they could be as exotic particles or substrates to induce nucleation. Therefore, this process can be termed inhomogeneous nucleation. The above substances were mixed thoroughly for 20 min using an automatic mixer. The homogeneous batch was transferred to a quartz crucible; then, we placed the crucible into a high-temperature furnace, preheated at 1373 K for 30 min. We raised the temperature to 1793 K at a rate of 10 K/min until the batch completely melted. To ensure homogeneity and clarity of the glass melt, we held the temperature at 1793 K for 3 h. Furthermore, the mold was preheated to 473 K. Subsequently, the molten glass was poured into the mold and quickly cooled to a glass block, which was transferred into a muffle furnace and annealed at 723 K for 4 h to relieve internal stresses.

The non-isothermal kinetic parameters of the crystals were determined using differential scanning calorimetry (DSC) [11]. About 30 mg of fine glass powder (1–10 μm) was used for the differential scanning calorimetry, which was analyzed using a simultaneous thermal analyzer (DSC/DTA-TG) NETZSCH STA 449 F5. The powders were packed in a quartz crucible and heated at different rates (5 K/min, 10 K/min, 15 K/min, and 20 K/min). The area under the DSC peak was calculated according to a method detailed elsewhere [11].

High-temperature X-ray diffraction (HT-XRD) was used to determine the kinetic parameters of crystalline phase transitions and precipitation. In this paper, an in situ HT-XRD study was conducted using the Bruker Instruments D8 Advance high-temperature X-ray diffractometer. The sample was heated from room temperature to 1213 K at a rate of 10 K/min. When the temperature rose to 833 K, XRD diffraction traces in the 2θ range of 10° to 80° were collected once every 20 K increase in temperature. The scanning speed was 10°/min. Phase quantification and crystallite strain analysis were carried out using TOPAS software with a full-pattern Rietveld refinement.

The crystallization process of Li_2_O–Al_2_O_3_–SiO_2_ glass was observed using a confocal scanning laser microscope (CSLM) (VL2000DX Laser Technology, Lasertec, Japan). The CSLM technique can record high-temperature transient phenomena and give clear three-dimensional (3D) contours at room temperature, representing an effective approach to track melting and solidification [19,20].

## 3. Results and Discussion

### 3.1. Differential Scanning Calorimetry (DSC)

The study of the crystallization kinetics of glass-ceramics is divided into isothermal and non-isothermal methods [11,12,13,14,15]. Due to the limitations of isothermal methods, non-isothermal methods have been mainly used for the research. During the study of crystallization kinetics, kinetic parameters, including crystallization activation energy, preexponential factors, and reaction models, can be used to analyze the crystallization mechanism [21]. Differential scanning calorimetry (DSC) is the most commonly used method for studying crystallization kinetics, as well as non-isothermal studies. The multi-rate scanning non-isothermal method is commonly used in research. Different heating rates have been used for the calculation of crystallization activation energy “*E_c_*”. In this paper, the heating rates were set to 5 K/min, 10 K/min, 15 K/min, and 20 K/min.

The DSC curves for the different heating rates of the Li_2_O–Al_2_O_3_–SiO_2_ glass-ceramics are shown in Figure 1. There are clearly two crystallization peaks on the DSC curve, indicating that the main crystalline phase of these glass-ceramics was a composite of two crystalline phases. The characteristic temperatures of the two precipitation peaks are shown in Table 1. The peak temperature of the two exothermic peaks shifted to a higher temperature with increasing heating rates. This shift in peak forms the basis for the determination of activation energy of crystallization using the linear and nonlinear model fitting methods of Kissinger and Kissinger–Akahira–Sunose, as discussed below.

### 3.2. Kine

#### 3.2.1. Kissinger Method

When using differential thermal analysis (DTA) or differential scanning calorimetry (DSC), the Kissinger method assumes that the reaction rate d*α*/d*t* is maximum at temperature *T_p_*, and that the conversion rate *α* is within a certain temperature range at different heating rates *β*, with a constant value *α_p_*. On the basis of these assumptions, the following expression for the relationship linking *T_p_*, *β*, and *E_c_* is established [22,23]:(1)$$\ln\left(\frac{T_{p}^{2}}{\beta }\right)=\frac{E_{c}}{RT_{p}}+C$$

The *T_p_* value can be determined from the DSC curve, and the calculation of the *E_c_* value can be performed using the above equation. The two crystallization peaks were analyzed separately according to the DSC curve shown in Figure 2. *E_c_* can be calculated from the slope of the ln(*T_p_*^2^/*β*)~1000/*T_p_* straight line. The resulting *E_c_* value calculated for the first peak was 381.03 ± 15.21 kJ/mol, while that for the second peak was 560.95 ± 15 kJ/mol. The error value when fitting the two straight lines was within the allowable standard error range. The value is close to those reported for Li_2_O–Al_2_O_3_–SiO_2_ glass ceramics, ranging from 125 to 476 kJ/mol [13,14,24]. The activation energy value are close to the bond strengths of Li–O, Mg–O, Al–O, and Si–O which are 326, 331, 485, and 700 kJ/mol [25], respectively, suggesting that crystallization may be controlled by the breaking and formation of bonds.

In addition, according to the basic part of the crystallization kinetics, there was a correlation between “*E_c_*” and the Avrami index “*n*”. The Avrami index “*n*” was calculated as follows [26]:(2)$$n=\frac{2.5}{\Delta T}*\frac{RT_{p}^{2}}{E_{c}}$$
where “Δ*T*” is the half-peak width of the crystallization peak of the differential scanning calorimetric (DSC) curve. Thus, the Avrami index “*n*” for the first peak at different heating rates was 2.67 (5 K/min), 3.20 (10 K/min), 4.79 (15 K/min), and 6.19 (20 K/min). The Avrami index “*n*” for the second peak was 2.95 (5 K/min), 3.88 (10 K/min), 5.69 (15 K/min), and 6.74 (20 K/min).

#### 3.2.2. Kissinger–Akahira–Sunose (KAS) Method

In the JMAK model, “*n*” and “*E_c_*” are considered to be constant values throughout the crystallization process [27]. With the development of related research, it was found that the change in nucleation and crystal growth rate could lead to changes in “*n*” and “*E_c_*” values of glasses with different crystallization degrees, regardless of the isothermal model (32 = 27 + 5) or non-isothermal model was used [28].

The crystalline phase transition fraction (*α*) is an important parameter in the thermodynamic analysis method. According to the DSC curve, the volume fraction of crystal phase transition can be estimated by calculating the area of exothermic peak [29]. In Figure 3a, “*S*” is the total area of the crystallization peak, and “*S_t_*” denotes the area of the crystallization exothermic peak when the temperature is “*T*”. The equation of the KAS method is expressed as follows (see plots in Figure 3b,c):(3)$$\ln\left(\frac{T_{\alpha }^{2}}{\beta }\right)=\frac{E_{c\left(\alpha \right)}}{RT_{p}}+C$$
where “*T_α_*” is the temperature corresponding to different degrees of crystallinity, and “*α*” is the crystallization volume fraction. The relationship between ln(*T_α_*^2^/*β*) and 1000/*T_α_* for the first and second crystallization peaks as a function of *α* is shown in Figure 4a,b. The crystallization activation energy *E_c_*(*α*) for the first and second crystallization peaks cannot be obtained from the fitted lines.

The relationship between *n_c_*(*α*) and *E_c_*(*α*) can be expressed as follows [30,31]:(4)$$n_{c(\alpha )}=-\frac{R}{E_{c\left(\alpha \right)}}*\frac{\partial \ln\left[-\ln\left(1-\alpha \right)\right]}{\partial \left(1/T\right)}$$

The relationship between ln[−ln(1 − *α*)] and 1000/*T* for the first and second crystallization peaks at different rates of temperature rise is shown in Figure 4c,d.

In the KAS method, “*E_c_*” and “*n*” correspond to the activation energy and Avrami index for a given degree of crystallinity. As a result, *E_c_*(*α*) and *n_c_*(*α*) are also referred to as the local activation energy and the local Avrami index [28,32]. The relationship between *E_c_*(*α*) and *n_c_*(*α*) as a function of crystallinity fraction (*α*) for the first and second peaks is shown in Figure 4e,f, where the errors meet the standard error requirement for a linear fit. The *E_c_*(*α*) of the first peak was 416.78 ± 24.94 kJ/mol at the beginning of the crystallization process; then, with a continuous decrease in the crystallization process, it decreased to 341.46 ± 29.26 kJ/mol at the end of crystallization. On the other hand, the *E_c_*(*α*) of the second peak decreased from 590.29 ± 33.26 kJ/mol at the beginning of the crystallization process to 498.81 ± 22.28 KJ/mol at the end.

The crystallization activation energy *E_c_*(*α*) is composed of the nucleation activation energy “*E_n_*” and the crystal growth activation energy “*E_g_*” [33]. Their relationship can be expressed as follows:(5)$$E_{c\left(\alpha \right)}=aE_{n}+bE_{g}$$
where “*a*” and “*b*” represent the proportion of nucleation and crystal growth at a certain stage in the crystallization process, and the relationship between them can be expressed as *a* + *b* = 1. It is well known that, at the beginning of the crystallization process, the nucleation process is dominant. Assuming that *b* = 0 at this stage, *E_c_*(*α*) is considered to be close to the value of “*E_n_*”. In the later stages of the crystallization process, the crystal growth process (*a* = 0) takes the place of the nucleation process, where *E_c_*(*α*) can be approximated by *E_g_*. It can be seen from Figure 4a,b that, with the increase in crystallization integral *α* from 0.1 to 0.9, the crystallization activation energy *E_c_*(*α*) showed a downward trend. This is due to the fact that the activation energy *E_n_* required in the nucleation stage was much greater than the energy Eg required for crystal growth.

As shown in Figure 4e,f, *n_c_*(*α*) increased with the crystallization fraction. For the first peak, when crystallization started, *n_c_*(0.1) < 2, whereas, when the crystallization process was nearly completed, *n_c_*(0.9) >2.5. This indicated that the crystallization mechanism changed from surface crystallization to overall crystallization. The *n_cvalue_* of the second peak was always higher than 3, indicating that the crystallization process was overall crystallization at all timepoints.

### 3.3. HT-XRD

The DSC results show that the Li_2_O–Al_2_O_3_–SiO_2_ glass-ceramic was composed of two crystalline phases, with a more complex crystallization process. Therefore, HT-XRD was used to analyze the crystallization process, and the results are shown in Figure 5a. When the temperature increased to 853 K, a minimal number of (Al_0.9_Cr_0.1_)_2_O_3_ crystals precipitated (JCPDS #51-1394). Upon heating to 893 K, a thimbleful of Li_2_SiZn crystals (JCPDS #25-0497) precipitated. Li_2_Si_2_O_5_ crystals (JCPDS #40-0376) began to precipitate in great quantities at 933 K. When the temperature increased from 933 to 973 K, the crystalline phase that precipitated was dominated by Li_2_Si_2_O_5_. Subsequently, LiAlSi_4_O_10_ crystals (JCPDS #35-0463) precipitated at 1013 K, accompanied by a small amount of Li_2_Si_2_O_5_ phase. MDI JADE and TOPAS software analysis showed that the overall crystallinity of the glass was above 80% at a temperature of 1133 K. At this point, the ratio of the two crystals (Li_2_Si_2_O_5_:LiAlSi_4_O_10_) was close to 28.3:71.7.

It can be seen from the HT-XRD image that, with the increase in temperature, the crystallization integral number of Li_2_Si_2_O_5_ crystal increased from 0% to about 15%, whereas the crystallization integral number of LiAlSi_4_O_10_ increased from 0% to about 65%. The crystallization process was analyzed using the KAS method. The relationship between ln[−ln(1 − *α*)] and 1000/*T* for Li_2_Si_2_O_5_ and LiAlSi_4_O_10_ at different rates of temperature rise is shown in Figure 5c,d. The activation energy *E_c_*(*α*) of Li_2_Si_2_O_5_ decreased from 405.72 ± 58.20 kJ/mol to 366.91 ± 50.72 kJ/mol with the change in crystallization volume. The activation energy of LiAlSi_4_O_10_ decreased from 560.77 ± 55.12 kJ/mol to 516.91 ± 47.21 kJ/mol.

The HT-XRD results according to the KAS method were analyzed, revealing that, for Li_2_Si_2_O_5_ and LiAlSi_4_O_10_ crystals, their crystallization activation energy decreased with the increase in crystallization integral number. In the early stage of crystallization, nucleation energy was the main factor, and the energy required for the nucleation process was relatively high. Upon the nucleation process being completed, the energy was basically used for crystal growth. According to the analysis of the HT-XRD results, the average grain size was approximately 100 nm. This was due to the high nucleation energy in the early stage of crystallization, resulting in a large number of crystal nuclei and limited crystal growth space, eventually forming dense nanocrystalline glass-ceramics.

### 3.4. CLSM

In situ measurement of glass samples at high temperature was carried out using a confocal scanning laser microscope (CSLM), and the changes in crystal size, morphology, and quantity were observed from room temperature to 1233 K. Figure 6 shows an image of the crystallization process of Li_2_O–Al_2_O_3_–SiO_2_ glass taken using a confocal scanning laser microscope (CLSM). In order to analyze the variation of grain number and size with temperature, the number of grains of different sizes at each temperature was counted, as shown in Figure 7.

As shown in Figure 6a, when the temperature was below 923 K, no significant change could be observed from the image, whereby only the existence of glass matrix could be seen. When the temperature was raised to 953 K, a small number of tiny nearly spherical crystals appeared, as shown in Figure 6b. On the basis of the DSC and HT-XRD results, it was presumed that these may be Li_2_Si_2_O_5_ crystals with a crystal size of approximately 50–70 nm. As shown in Figure 6c, when the temperature was increased to 1013 K, a large number of LiAlSi_4_O_10_ spherical crystals appeared with a particle size distribution in the range of 50–90 nm. As the temperature continued to rise, the number of crystals gradually increased, the number of grains in the range of 90–130 nm increased in quantity, and the glass matrix portion decreased obviously, as shown in Figure 6d–h. When the temperature was 1133 K, the percentage of grains in the 90–130 nm range was over 85%. Due to the large number of crystal nuclei, the growth space of crystal grains was limited. Once the grains were arranged densely and came into contact with each other, the grain size no longer grew with the increase in temperature. According to the results of the crystallization kinetic model, the crystallization mechanism of Li_2_O–Al_2_O_3_–SiO_2_ glass was the three-dimensional growth of crystals. The analysis of the HT-XRD results and the CLSM images showed that the Li_2_O–Al_2_O_3_–SiO_2_ glass formed a large number of crystal nuclei during the nucleation phase. The crystal nucleus constituted a large number of densely arranged nanoscale crystals. At the end of the growth process, the crystalline volume fraction exceeded 80%.

## 4. Conclusions

In this paper, DSC, HT-XRD, and CLSM were used to investigate the crystallization process of Li_2_O–Al_2_O_3_–SiO_2_ glass. There were two exothermic peaks at 943 K (the first peak) and 1009 K (the second peak) on the DSC curve, corresponding to the precipitation of Li_2_Si_2_O_5_ and LiAlSi_4_O_10_ crystals, respectively.

The DSC results were analyzed using the non-isothermal model. For the first peak, the crystallization mechanism changed from surface crystallization to overall crystallization; however, the second peak was overall crystallization throughout. The results of the activation energy calculations using the Kissinger method were 381.03 ± 15.21 kJ/mol for the first peak and 560.95 ± 15 kJ/mol for the second peak. Using the Kissinger–Akahira–Sunose (KAS) method, the activation energy at the beginning of the precipitation phase was 416.78 ± 24.94 kJ/mol for the first peak and 590.29 ± 33.26 kJ/mol for the second peak. Furthermore, the activation energy continued to decrease as the crystals continued to precipitate, ending at 341.46 ± 29.26 kJ/mol for the first peak and 498.81 ± 22.28 kJ/mol for the second peak. During this process, the Avrami index (*n_c_*) for both crystal precipitation processes increased with *α*.

According to the HT-XRD results and CLSM images analyzed, the main crystalline phases were LiAlSi_4_O_10_ and Li_2_Si_2_O_5_. Analysis of the HT-XRD results using a non-isothermal model, i.e., the KAS method, revealed that nucleation was the main process at the beginning of crystallization. After nucleation, energy was mainly used for crystal growth. During this process, the activation energy of crystallization decreased with the increase in the integral number of crystallization. Only a small number of Li_2_Si_2_O_5_ crystals precipitated at the beginning of the crystallization, and the crystal size was about 50–70 nm. Then, LiAlSi_4_O_10_ started to precipitate, forming LiAlSi_4_O_10_ and Li_2_Si_2_O_5_ composite spherical crystals. The size of the grains at the end of the precipitation phase was mainly in the range of 90–130 nm. According to analyses of the crystallization kinetics and crystallization process, the crystallization process control of Li_2_O–Al_2_O_3_–SiO_2_ glass could be established.

## Figures and Tables

**Figure 1 materials-15-08006-f001:**
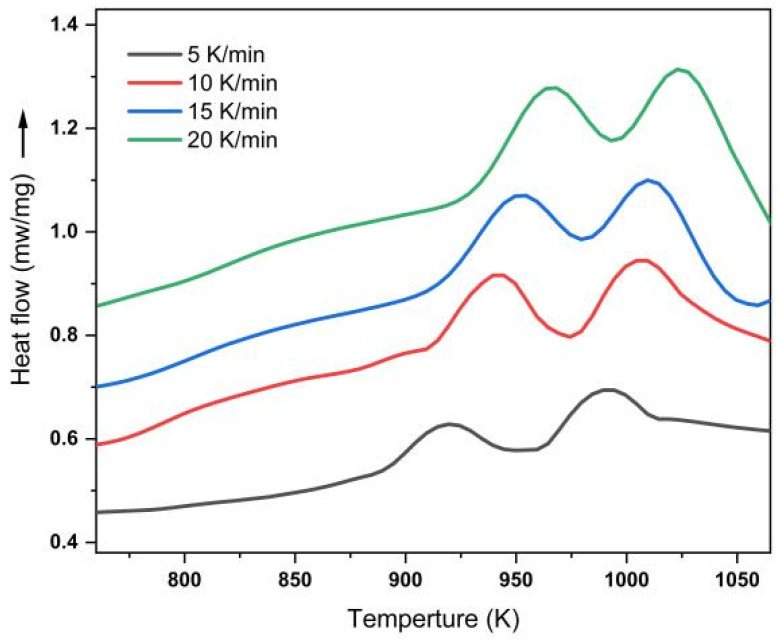
DSC curves of Li_2_O–Al_2_O_3_–SiO_2_ glass samples at different heating rates.

**Figure 2 materials-15-08006-f002:**
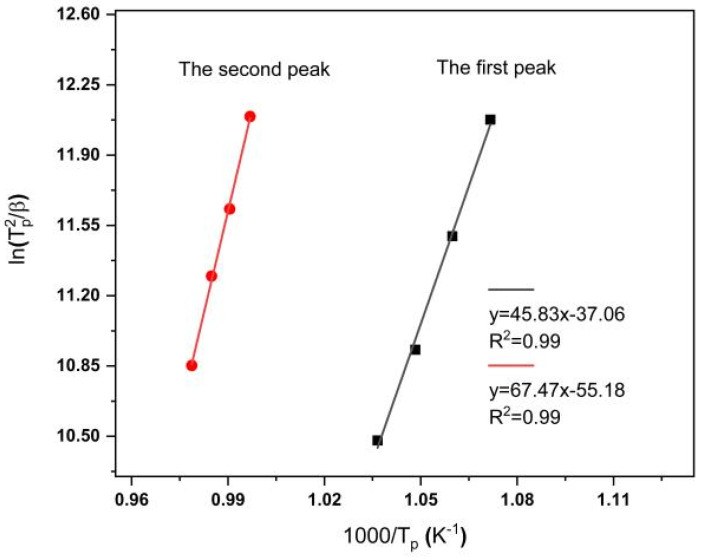
Schematic diagram of the relationship between ln(*T_p_*^2^/*β*) and 1000/*T_p_* for analysis of the first peak and the second peak.

**Figure 3 materials-15-08006-f003:**
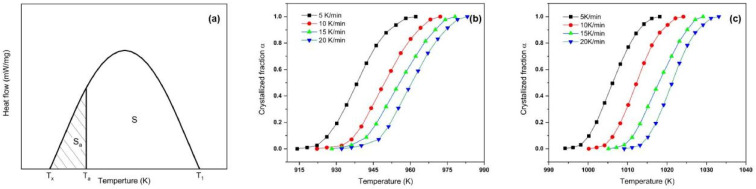
(**a**) Schematic diagram of the determination of the volume fraction of the crystalline phase transition; (**b**,**c**) variation of crystal integral number under different heating rates of the first and second crystallization peaks.

**Figure 4 materials-15-08006-f004:**
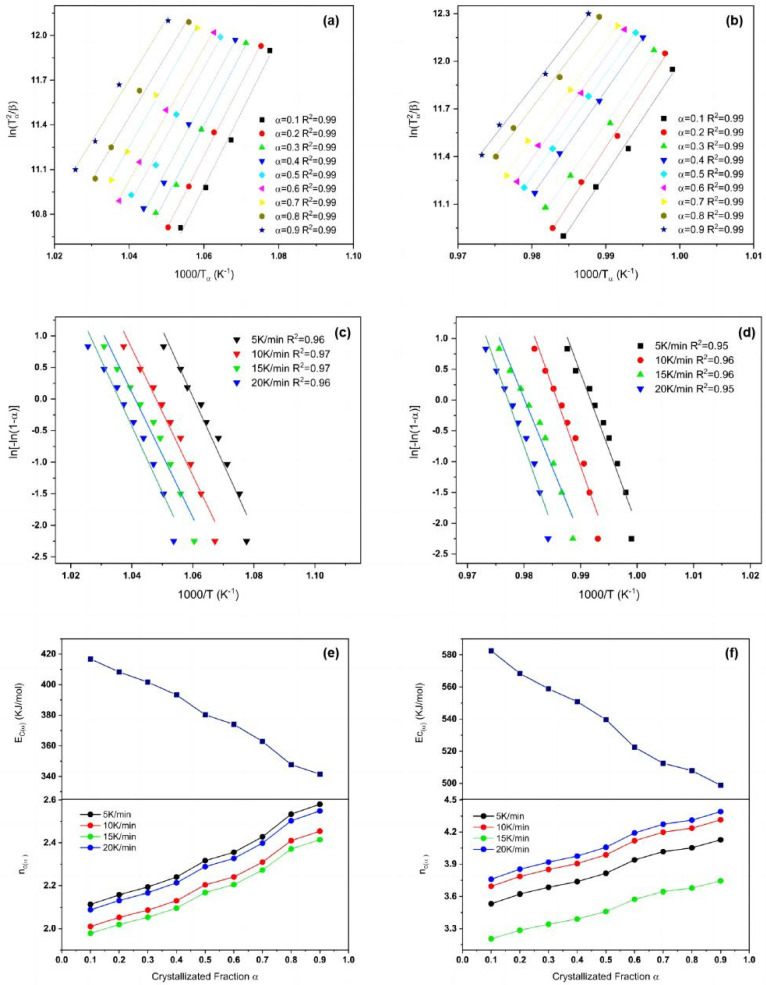
(**a**,**b**) Plots of ln(*T_α_*^2^/*β*) vs. 1000/*T_α_* for different integration numbers of the first and second crystallization peaks; (**c**,**d**) plots of ln[−ln(1 − *α*)] vs. 1000/*T* for different precipitation integration numbers of the first and second crystallization peaks; (**e**,**f**) *E_c_*(*α*) and *n_c_*(*α*) as a function of crystallized fraction *α*; the error bars correspond to the standard errors from linear fitting.

**Figure 5 materials-15-08006-f005:**
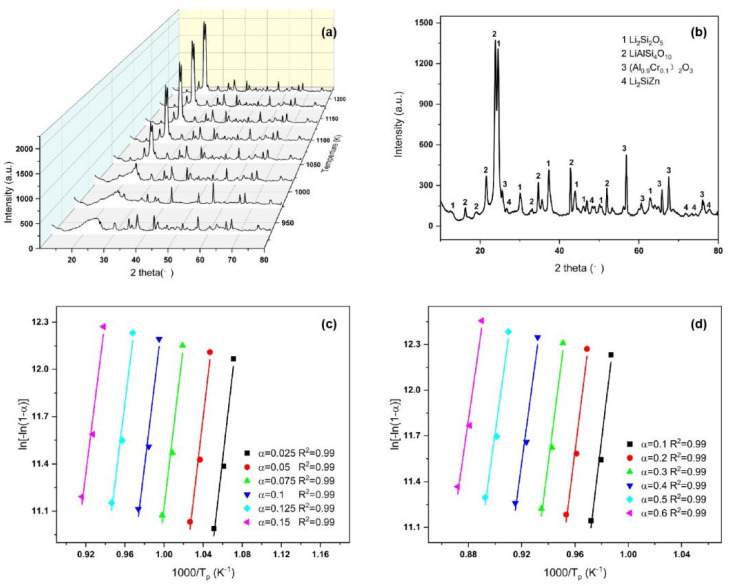
(**a**) HT-XRD patterns of Li_2_O–Al_2_O_3_–SiO_2_ glass samples from 833 K to 1213 K; (**b**) XRD patterns of L_i2_O–Al_2_O_3_–SiO_2_ glass samples at 1133 K; (**c**,**d**) plots of ln(*T_α_*2/*β*) vs. 1000/*T_α_* for different integration numbers of Li_2_Si_2_O_5_ and LiAlSi_4_O_10_.

**Figure 6 materials-15-08006-f006:**
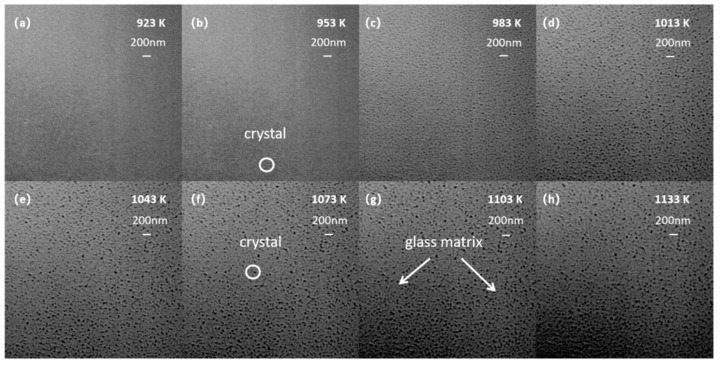
Images of Li_2_O–Al_2_O_3_–SiO_2_ glass crystallization process. (**a**) Crystallization images at 923K; (**b**) Crystallization images at 953K; (**c**) Crystallization images at 983K; (**d**) Crystallization images at 1013K; (**e**) Crystallization images at 1043K; (**f**) Crystallization images at 1073K; (**g**) Crystallization images at 1103K; (**h**) Crystallization images at 1133K.

**Figure 7 materials-15-08006-f007:**
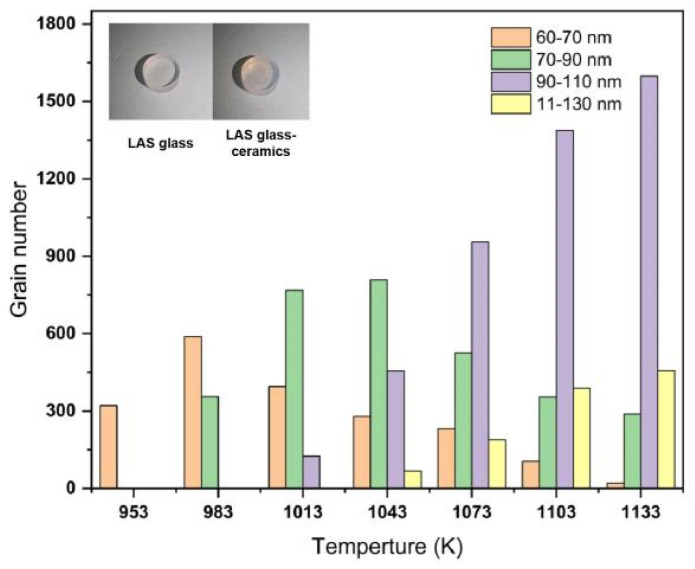
Grain size and number at different temperatures.

**Table 1 materials-15-08006-t001:** Characteristic temperatures measured using non-isothermal differential scanning calorimetry.

Temperature (K)		The First Crystallization Peak		The Second Crystallization Peak	
Heating Rate (K/min)	*T_g_*	*T_c_* _1_	*T_pc_* _1_	*T_c_* _2_	*T_pc_* _2_
5	789	918	934	994	1003
10	798	929	943	1000	1009
15	807	937	954	1006	1015
20	815	948	965	1011	1022

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
