# Peer review of "Study on Crystallization Process of Li2O–Al2O3–SiO2 Glass-Ceramics Based on In Situ Analysis"

_materials, 2022, doi:10.3390/ma15228006_

Round 1

Reviewer 1 Report

The paper labelled « Study on crystallization process of Li2O-AL2O3-SiO2 glass ceramics based on in-situ analysis» by Minghan Li and co-workers details a high temperature investigation of the crystallization processes of petalite and lithium disilicate through DSC, HT-XRD and CSLM analyses. The activation energy and Avrami index of both crystallizations were also calculated by Kissinger and KAS method.   Data presented are clear and consistent and, regarding the objectives and policies of the journal, publication could be envisaged in Materials. However, despite the quality of the present article, minor points should be considered before publication:

-        The formula of Kissinger plot is not correct; actually the authors should plot ln(beta/Tp2) versus 1000/Tp  (and not ln(Tp2/beta). Then the slope of the linear fit should be equal to -Ea/R. Please correct this mistake.

-          The overall quality of the figures should be improved, it is sometimes difficult to clearly see the labels on them (resolution should be increased); errors bars are claimed on figure 4 but I do not see any error bar. The most problematic pictures are in figure 6, it is highly difficult to see sharp images, the scale of the pictures or the crystals themselves… Moreover, writing in red over dark images does not help to visualize correctly.

-          It would have been appreciated, in the discussion part, to compare the results of the manuscript with existing literature (Kumar, Huidong Li, Lilensten) and especially point out the benefits from the current work. The authors should put their results forward to highlight the new insights on the studied system. They noteworthy claim that the crystallization process control can be guided, but it would have been appreciated to detail more about it.

-          Line 221 “and crystallization process was more complex”, apart from the fact that syntax is not correct (“more complex” is waiting from “than something”), I do not understand what the authors stated by that. Is it because two different crystal phases appear that the process is more complex (than expected?) ? It is unclear to me, please explain it.

Few typos errors remained:

-          In the abstract, it is not necessary to give twice the formula of petalite and lithium disilicate.

-          In the main text, it would be better to refer to equations and figures by calling them directly instead of writing “as follows”

-          Line 100: The powders… “were” instead of “was”

-          Line 135 (figure caption of Table 1): use “the first and second crystallization peaks”

-          On Table 1, “onset temperature” would be rather preferred than “starting value”

-          Line 143: even though it could appear obvious, Ec is not defined as the activation energy (moreover, the latter was identified as “E” on line 126)

-          Line 157: please used “crystallization peak” or “exothermic peak” instead of “crystalline peak”

-          Line 195: a ± is missing between 341.46 and 29.26

-          Line 270: there is an unnecessary space between “w” and “hen”

-          Line 288: there are two points at the end of the sentence.

-          Line 289-290 “it was found that the crystallization… to whole crystallization” : the syntax of the sentence is not correct. Please rephrase it.

-          Line 308: as the subject of the sentence is “the size of the grains”, please use “was” instead of “were”

Reviewer 2 Report

the file is attached

Reviewer 3 Report

The submitted paper deals with crystallization phenomena of Li2O-Al2O3-SiO2-based glasses. A few remarks are directed to the Authors, namely:

-The first part of “Experimental” should be rewritten, because function of a few additional reagents (B2O3, P2O5, Na2CO3 or ZrO2) are not there described.

-Photos of as-melted glasses and samples after crystallization process can be included.

-In DSC study, glass transition temperature Tg, glass crystallization temperature Tc and the maximum crystallization peak temperature Tpc should be pointed and discusses more in details.

-The tendency to homogeneous or inhomogeneous nucleation processes in material under study may be described.

-Technical quality of the paper can be improved as for example, separation between values and units.

-Red color on Figure 6 is completely indistinct.

Round 2

Reviewer 3 Report

The reviewer's comments have been considered by Authors, hence this paper can be accepted for publication.